# Encryption and steganography of synthetic gene circuits

Oliver Purcell [1], Jerry Wang[1], Piro Siuti[1] & Timothy K. Lu[1,2,3]

Synthetic biologists use artificial gene circuits to control and engineer living cells. As engineered cells become increasingly commercialized, it will be desirable to protect the intellectual property contained in these circuits. Here, we introduce strategies to hide the design of synthetic gene circuits, making it more difficult for an unauthorized third party to determine circuit structure and function. We present two different approaches: the first uses encryption by overlapping uni-directional recombinase sites to scramble circuit topology and the second uses steganography by adding genes and interconnections to obscure circuit topology. We also discuss a third approach: to use synthetic genetic codes to mask the function of synthetic circuits. For each approach, we discuss relative strengths, weaknesses, and practicality of implementation, with the goal to inspire further research into this important and emerging area.

---

[1] Synthetic Biology Center, Massachusetts Institute of Technology, 500 Technology Square, Cambridge, MA 02139, USA. [2] Department of Biological Engineering, Massachusetts Institute of Technology, 77 Massachusetts Avenue, Cambridge, MA 02139, USA. [3] Research Laboratory of Electronics, Department of Electrical Engineering and Computer Science, Massachusetts Institute of Technology, 77 Massachusetts Avenue, Cambridge, MA 02139, USA. Correspondence and requests for materials should be addressed to T.K.L. (email: timlu@mit.edu)

A central goal of synthetic biology is to design and construct synthetic gene circuits that can be used to engineer novel functions into living organisms[1,2]. Successfully engineering synthetic gene circuits is resource and labor intensive and the corresponding intellectual property (IP) is valuable. However, DNA sequencing technology is now at a stage where the genome of an engineered cell can be sequenced cheaply and quickly, thus allowing the designs of synthetic circuits or pathways performing the critical functions to be readily determined. New methods to protect the IP contained within these organisms, beyond legal remedies, are therefore needed. While this is a recognized problem, to date there has only been limited research into possible solutions[3,4].

Here, we present two methods for hiding the design of synthetic genetic circuits: encryption of the circuit design by scrambling circuit topology using uni-directional recombinases ("circuit scrambling") and obscuring circuit topology using additional circuit "dummy" components ("circuit camouflage"), an application of steganography. We also briefly discuss masking circuit function using synthetic genetic codes ("circuit re-encoding"), which is still in early stages of feasibility but has potential utility. Here, topology is defined by functional interactions between components in a genetic circuit. The aim of these approaches is to make it more difficult for unauthorized third parties to uncover the structure and function of a given artificial gene circuit. We demonstrate experimental proof-of-concepts for both recombinase scrambling and circuit camouflage, and discuss circuit re-encoding, which remains technically challenging.

## Results

**Circuit scrambling**. Circuit scrambling is an approach to encrypt the topology of a genetic circuit, either in vitro or in vivo (i.e., within an engineered cell), which would offer protection when the circuit is being stored or transferred between parties. Circuit scrambling can be achieved through the use of site-specific uni-directional recombinases, such as large serine recombinases, that recognize specific DNA sequences known as *attB* and *attP* sites[5]. If these recombinase-recognition sites (RRS) are placed on the same piece of DNA, the cognate recombinase will cause a one-time recombination event between the RRS, resulting in inversion or excision of the DNA between the RRS, depending on their relative orientation[6,7]. Such recombination events can then be used to scramble the topology of a circuit when it is not in use, which is useful since the behavior of a gene circuit is largely determined by its topology and the biochemical properties of the components.

To demonstrate circuit scrambling using recombinases, we use transcriptional circuits as an example. Here, the topology of transcriptional circuits is determined by specific promoter-gene pairings, where promoters express genes encoding transcription factors, and each promoter can be regulated by one or more transcription factors. A process that can deterministically "scramble" and "unscramble" these pairings would allow for encryption and decryption of the circuit topology, resulting in non-functional or functional circuits, respectively.

Uni-directional recombinases provide a means of deterministically scrambling and unscrambling circuit topology by physically re-structuring the DNA. Figure 1 illustrates a proof-of-concept of the encryption process for a genetic AND gate. The AND gate is comprised of three genes, organized linearly on a stretch of DNA (Fig. 1a). The gate is scrambled using an iterative two-step process: (1) a section of DNA is chosen such that at least one end is between a promoter and its gene and that section is inverted, and (2) RRS flanking the inverted region are introduced into the sequence. Multiple sets of RRS can be introduced in this

way to scramble the circuit (Fig. 1b). The scrambled circuit can then be synthesized or assembled in vitro. Unscrambling (Fig. 2a) requires applying the same set of the recombinases that were used for the scrambling process in a restricted set of orderings (an analysis of this for the AND gate is discussed later). By overlapping pairs of recombinase sites corresponding to different recombinases, order dependency is introduced into the unscrambling process. The order-dependent application of uni-directional recombinases has been used previously in the construction of genetic logic circuits[1,8], while bi-directional recombinases have been used to "decompress" a single genetic circuit structure into an equilibrium of more than one structure[9].

Circuit scrambling can be substantially strengthened using decoys. Decoys are genetic elements that are present within the scrambled construct, and could conceivably be a part of functional circuit, but are not actually required for the circuit's function. Decoys introduce additional uncertainty as to what the actual topology of the functional circuit is, making cracking the scrambled circuit harder. The possibility of decoy genes permits excision events to be incorporated into the unscrambling process, thus preventing excision from being an indication of incorrect unscrambling. The AND gate scrambling example illustrates the use of a decoy promoter-gene pair and decoy RRS (Fig. 1b). The decoy promoter-gene pair pD and ORF D corresponds to pLtet0-1 and *tetR*, which could be plausible elements of the functional circuit. The decoy RRS are those that are recognized by recombinase PhiC31. In this example, recombination with PhiC31 is not needed for the correct unscrambling, and if used would result in a deletion of the majority of the circuit components, leaving only the ORFs *tetR* and *gfp* remaining. The resulting circuit would not perform the correct AND gate function.

We experimentally implemented an example of circuit scrambling and unscrambling in Figs. 1, 2 in vitro. It should also be possible to carry out this process on circuits encoded within cells[1,10,11]. Starting with a plasmid containing the scrambled construct, we unscrambled the construct through successive rounds of in vitro treatment with different purified recombinases. At each round, plasmids were transformed into *E. coli* and recombined constructs were selected. In future work, this protocol could be optimized so that successive recombination, transformation, and selection events would not be necessary.

Brute-force cracking of recombinase-scrambled constructs could be achieved either by (1) constructing all possible circuits based on the promoters and genes present, or (2) trying all possible orders and identities of the recombinases. In either case, the identity of the true circuit must still be established from the resulting collection of potential topologies. . Additionally, decoys mean that a third party must estimate which elements are actual circuit components to avoid a final circuit with incorrect components.

A more systematic approach to cracking the scrambled circuit is to enumerate all possible DNA states obtainable from the different orderings of the recombinases, and then evaluate the states for how likely they are to be the true circuit. We enumerated all possible recombinase decryption orders ($\sum_{k=1}^{k=4} \frac{n!}{(n-k)!} = 64$)[12], where $n$ is the total number of recombinases and $k$ is the number of recombinases used in a possible decryption, for the encrypted AND gate (Supplementary Figure 1). It has been formally proven elsewhere that when using a single RRS pair per recombinase, $n$ recombinases can produce at most $2^n$ unique DNA states[11] (including the state where no recombinases are used). Furthermore, different orderings of the same set of recombinases can lead to the same DNA state[11]. In practice, not all orderings of recombinases are productive

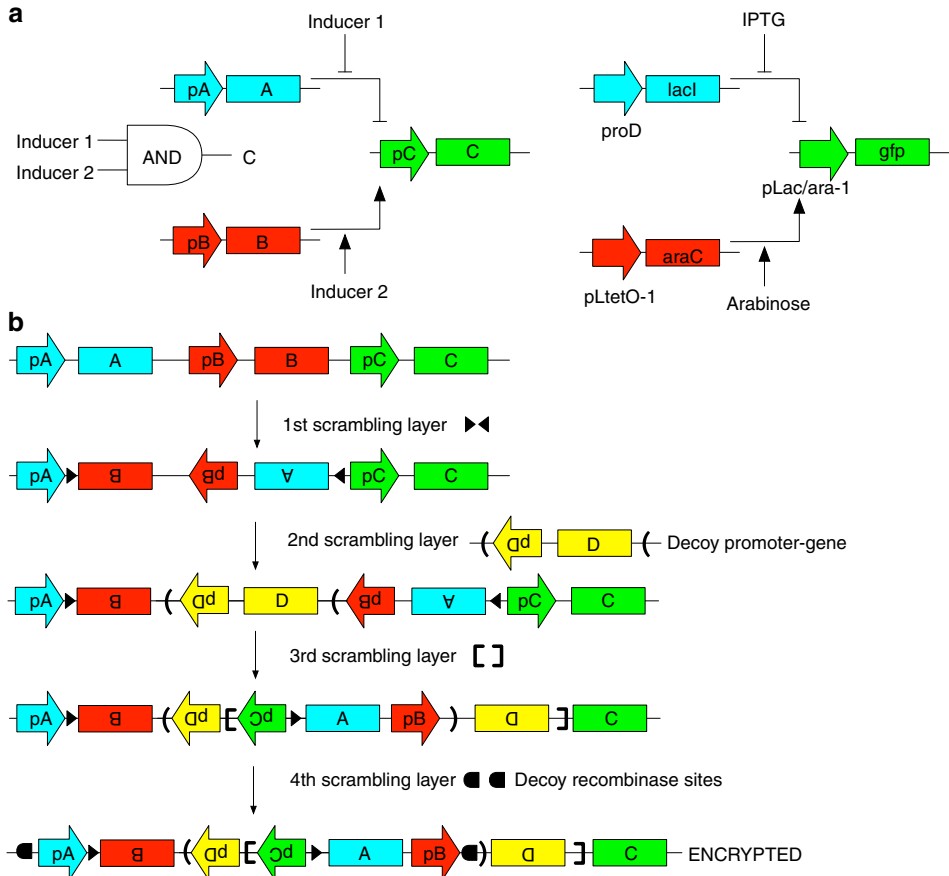

**Fig. 1** Encryption of a genetic AND gate using circuit scrambling. **a** Genetic AND gate. A, B, C ORFs encoding *lacI*, *araC*, and *gfp*, respectively. pA, pB, and pC are the promoters proD, pLtetO-1, and pLac/ara-1, respectively. Inducer 1 (IPTG) inhibits transcriptional repression by LacI, and Inducer 2 (arabinose) activates transcription by AraC. This circuit was tested in cells that do not express the TetR repressor so that pLtetO-1 is constitutively active. **b** Encryption of the AND gate. At each step, either a new set of recombination sites (decoy or not) or decoy promoter-gene pairs is added. Promoter pD and ORF D are pLtetO-1 and *tetR*, respectively. Recombinase sites denoted by triangles, brackets, square brackets, and semi-circles are recognized by recombinases TP01, BxbI, Int3, and PhiC31, respectively

because an excision event may remove other recombination sites, which cannot then be used in subsequent steps. However, because any ordering of the same set of recombinases leads to the same DNA state, all orderings have to be unavailable for the state to be unreachable. For the AND gate, $n = 4$ and $2^n = 16$, and although excision events only permit 35 of the possible 64 recombinase orderings to have a productive effect (one where every recombinase in the set has an effect on the DNA state) on the DNA state when performed (Supplementary Figure 1), we find that all 16 states are reachable. Two of these states are identical, owing to nested excision events. Due to occurrences of this type of redundancy, while the number of unique DNA states that can be reached may typically be less than $2^n$, it may not be subtantially less.

Thus, an attacker would be presented with a list of possible unique circuit candidates, which even if only numbering in the low hundreds (e.g., $2^8 = 256$) may include many plausible candidates. In the case of the AND gate, despite its small size, many of the 15 DNA states (16 including the encrypted state) are plausible configurations of promoter-ORF pairings and circuits (supplementary Figure 1). Further examination of these configurations suggests ways in which to increase the diversity and the number of connections in these incorrect circuits, thus making them appear more plausible. For instance, using different variants of pLtetO-1 for pB and pD would differentiate between some configurations, adding a promoter permanently (i.e., with no

recombinases sites between the promoter and ORF so that this relationship is never broken) driving *tetR* expression would introduce *tetR* links into many configurations, and either making pC bi-directional or more simply adding in an opposing promoter would in many cases express *lacI* and introduce a feedback loop. None of these additions would affect the topology of the correct decrypted AND gate, although there could be a quantitative effect of making pC bi-directional. Finally, in this example, the correct AND gate circuit is far simpler (e.g., contains no unused promoters or ORFs, and has a single promoter per gene, with all promoter-gene pairings in the same direction) than many of the other candidate states. In practice, the correct circuit should be designed to look comparably unstructured as the other candidates so that it does not stand out.

Practical implementation of this approach is limited by the number, orthogonality, and efficiencies of available recombinases. For example, 11 recombinases have been demonstrated to be largely highly orthogonal to each other[13], and more could be discovered through further mining. Three of the recombinases display low levels of cross-talk with a single other recognition site in addition to their own. The orthogonality of the set of recombinases used is therefore important as cross-talk would lead to incorrect recombination events. If necessary cross-talk could likely be reduced by engineering and directed evolution. Recombinase efficiencies can affect the time required for successful decryption, as well as the overall decryption efficiency.

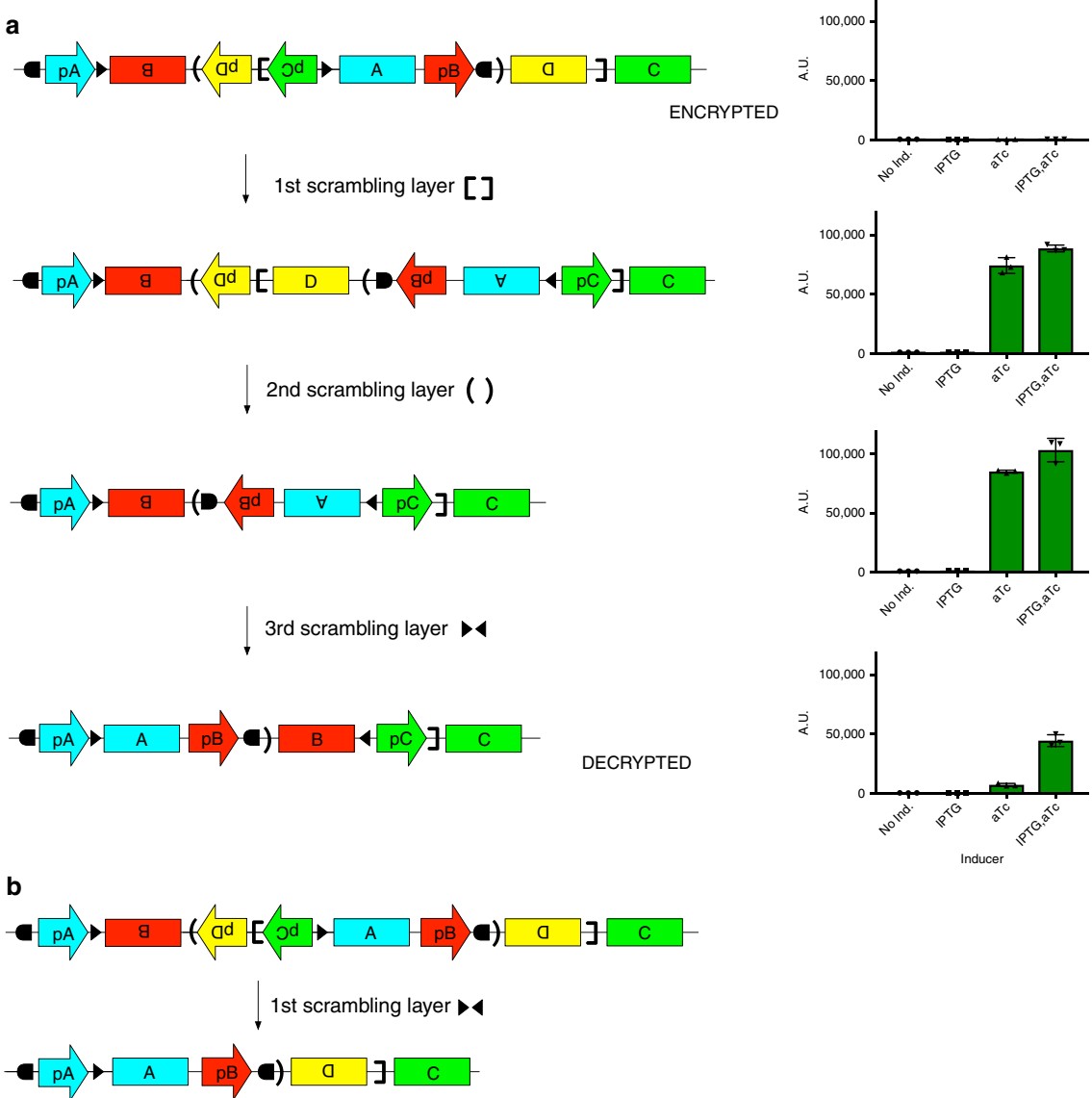

**Fig. 2** Decryption of the genetic AND gate. **a** Unscrambling of the scrambled AND gate from Fig. 1. Behavior of constructs from each stage of the unscrambling is shown. $N = 3$, error bars represent standard deviation. The final unscrambled construct (step 4) demonstrates AND gate behavior, whereas intermediate steps show non-AND gate behavior. **b** An example of an incorrect unscrambling. RiboJ sequences[34] were used directly upstream of the RBS of each gene to prevent any interference from the increased size of the 5′UTRs (RiboJ54, RiboJ51, RiboJ53 and RiboJ, for *araC*, *lacI*, *tetR*, and *gfp* respectively). NoInd = no inducer, Arab = arabinose

For example, Bxb1 has been shown to recombine 90% of sites in 2 h in vitro[14]. Assuming this rate holds for other recombinases, a decryption using five recombinases would be expected to have a combined efficiency of $0.9^5 \sim 60\%$. Allowing recombinase reactions to run longer or performing evolution on the recombinases to improve their activity could help to mitigate these problems.

The recombinase reactions described here are uni-directional and do not permit reversibility unless they are used together with recombinase directionality factors[15]. This is acceptable for current applications, as engineered cells are often single-use and are rarely retrieved and used again. In future applications this may change, and thus re-scrambling may be useful. An example may be a field application outside of a research laboratory, where maintaining a bank of frozen stocks for long periods of time is not possible. It would then be useful to maintain a single bank of cells for an extended period of time, unscrambling the circuit when the cells need to be used, and then re-scrambling the circuit

afterwards. In addition, re-scrambling is also interesting as it poses technical and conceptual challenges. Next, we introduce an approach to obscuring circuits that is reversible.

**Circuit camouflage**. Unlike circuit scrambling, circuit camouflage maintains the topology of the true circuit, but makes this topology hard to determine. This is achieved by embedding the functional circuit within a larger "camouflaging" circuit (Fig. 3), a form of steganography. A similar strategy is employed in integrated circuit (IC) design[16–18], whereby additional dummy contacts between conducting layers are added so from the top-view (the view from which microscopy can be used to uncover the circuit design) the design of the circuit is not easily identifiable. In our approach de-camouflaging uses a molecular "key" to subtract the effects of the camouflaging genes from the functional circuit so that it can operate properly. In the implementation discussed here, this is achieved by repressing the expression of the

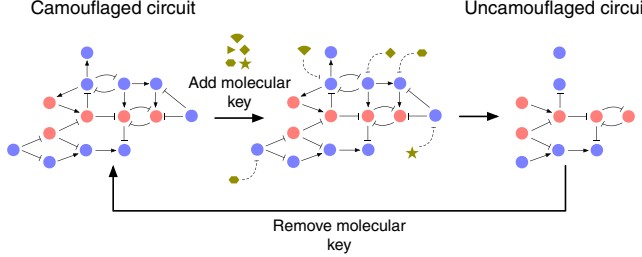

Camouflaged circuit Uncamouflaged circuit

Add molecular key

Remove molecular key

**Fig. 3** Circuit camouflage. The red nodes denote genes of the correct circuit while the blue nodes are the genes of the camouflaging circuit. The addition of a "molecular key" represses or removes the effects of genes of the camouflaging circuit that are incident on the correct circuit, leaving the correct circuit to function. Only nodes repressed by the molecular key and that are only incident on the correct circuit have been removed in the uncamouflaged circuit. Nodes targeted by the key but which still have an incident node from the correct circuit have been kept, as these nodes will act as a sink for the proteins from the correct circuit

camouflaging genes. As most repression mechanisms are reversible, de-camouflaging can be transient and the circuit can be re-camouflaged by removing the repressive modalities.

Figure 4 demonstrates an in vivo proof-of-concept implementation of circuit camouflage on a bi-stable switch circuit[19]. Figure 4a shows the bi-stable switch topology that relies on cross-repression by *tetR* and *lacI*, and its bi-stability in terms of a GFP output. To camouflage the circuit, we added two additional genes to the circuit, *araC* and *λCI* (Fig. 4b). By modifying the promoter driving *tetR* to include a binding site for *λCI*, *λCI* can bind to the promoters of the bi-stable switch, and in concert with activation of *λCI* expression by AraC, perturb its function by destroying the toggle switch behavior. To uncamouflage the circuit, we transformed in a plasmid containing constitutively expressed dCas9, along with two constitutively expressed guide RNAs targeting the ORFs of *araC* and *λCI*, which together form a "CRISPR key". The CRISPR key repressed expression of *araC* and *λCI* through CRISPR interference, leaving the uncamouflaged and functioning bi-stable switch (Fig. 4b). The plasmid contained the temperature-sensitive origin repA101[ts] from pDK46, which is stable at 30 °C but unstable at 42 °C, and the ampicillin resistance marker. By growing the cells on solid media at 42 °C overnight without selection, the plasmid was removed, thus re-camouflaging the circuit by making it lose its functionality due to the effects of the camouflaging genes.

Cracking the camouflaging by brute force requires finding all possible sub-circuits of the camouflaged circuit and determining the correct one. Using a molecular key that can target specific genes, the number of possible sub-circuits is $2^n$ where $n$ is the total number of genes within the camouflaged circuit. This is a numerically comparable scheme to the number of unscrambling routes for circuit scrambling. However, deciding which of these $2^n$ circuits is the true circuit is likely more difficult than with circuit scrambling. This is because promoters and ORFs are always paired in circuit camouflage, in comparison to circuit scrambling where both the promoter-ORF combination that expresses the transcription factor and the promoter-ORF that the transcription factor regulates have to be paired for the link to be plausible. This means that the universe of plausible circuit candidates from circuit camouflage will be larger and more highly connected on average than from circuit scrambling. If specific links can be targeted instead of only specific genes, the camouflaging can be strengthened, as there are typically more regulatory links than genes. There is no known simple mechanism that can discriminate between regulatory links from the same genes, but differentially blocking transcription factor

access to different promoters may be one strategy. For example, targeting a unique site that overlaps an activator-binding site could allow blocking of the activator binding to one promoter that contains the unique site but not another that does not contain the unique site. The same approach could work for eukaryotic repressors binding upstream of the core promoter.

For an ideal camouflaging scheme, the circuit should not display the correct qualitative and quantitative behavior when an incorrect key is used. We examined the behavior of our circuit under incorrect keys to understand the different classes of behaviors that might result in a general case. In Fig. 5, the number of possible gRNA targets (anywhere there is an NGG for *S. pyogenes* dCas9) is given below each promoter and ORF. The numbers for promoters are exact, while for ORFs they are estimated to be the length of the ORF in bp divided by 16 (the random probability of getting NGG in a triplet (CCN, which corresponds to NGG in the bottom strand), was not considered). The total number is 235, giving a total number of gRNA combinations of $2^{235} \sim 10^{70}$. While each of the $10^{70}$ combinations could be a potential key, for tractability we considered only one gRNA for each ORF as a potential key member. For the promoters, we considered only gRNAs that overlap with the core promoter region, and chose all of them to overlap the -10 region. This gives a set of 7 gRNAs that may form part of the correct key, labeled a-g on Fig. 3, for a total of $2^7 = 128$ combinations, or 127 if the no-gRNA combination is excluded. As a further simplification, we only examined combinations of at most 3 gRNAs, specifically a subset of these that illustrates the likely outcomes of using an incorrect key (Fig. 5). There were a range of qualitative and quantitative effects of using an incorrect key, when compared to the correct switch behavior (Fig. 4b, panel 2): The combinations a + b, a + b + g, and e + b gave outputs that were qualitatively different than switch behavior. Combinations c and c + d resulted in quantitatively different switch or switch-like behavior. Combination b + f yielded comparable switch behavior. Combination e + f resulted in comparable switch behavior but unreliably (not all repeats performed the same, also observed for a + b). These four behaviors: (1) no practical difference from the encrypted circuit (b + f), (2) quantitatively different from the correctly decrypted circuit (c and c + d), (3) qualitatively different from the correctly decrypted circuit (a + b, a + f + g, and e + b), and (4) quantitatively equivalent to the correctly decrypted circuit but inconsistent (e + f) cover the possible scenarios that could be seen in the general case.

Although most of the circuits that have been described in literature are proof-of-concept systems, their eventual use will be in the precise control of engineered cells and organisms, where the quantitative relationship between input and output is vital. Examples include cell classifiers for cancer[20] where false negative and positive rates are determined by quantitative circuit behavior, and engineered probiotics that sense biomarker levels and titrate the expression of therapeutic compounds to specific doses. Therefore, hiding the quantitative behavior of a circuit may be just as important as hiding the qualitative behavior.

While the topology of the correct circuit is destroyed in circuit scrambling, the topology can remain intact with circuit camouflage. The ratio of camouflage genes to true circuit genes is therefore important—the lower this ratio, the more likely that randomly selecting a connected sub-circuit will give you a part of the true circuit. As with circuit scrambling, circuit camouflage does not offer protection when the circuit has been uncamouflaged and is in use, although the point of weakness differs. For circuit scrambling, the vulnerability is the ability to sequence the unscrambled construct, while with circuit camouflage the vulnerability is the ability to obtain the molecular key when it is in the cell. Using other types of molecular keys that are either

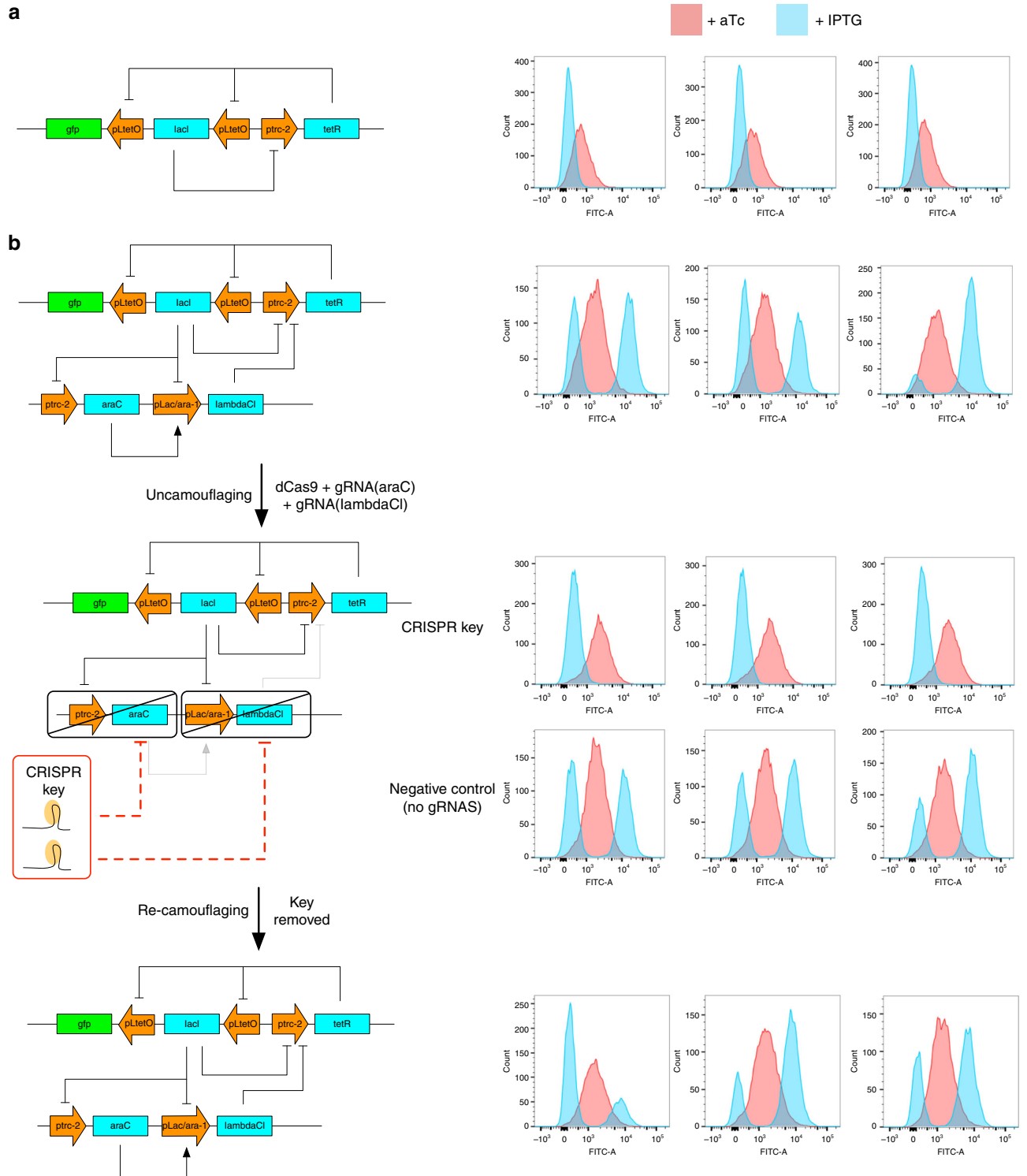

**Fig. 4** Camouflaging and uncamouflaging of a bi-stable switch using circuit camouflage. **a** The bi-stable switch is formed from *tetR* and *lacI* repressing each other. GFP serves as a reporter. **b** The additional genes *araC* and *λCI* interconnect with the circuit and camouflage it by changing the topology and interfering with its function. The addition of the CRISPR key represses expression of *araC* and *λCI*, uncamouflaging the switch. The dynamics of the uncamouflaged switch are comparable to the circuit without the addition of camouflaging genes. Removal of the CRISPR key re-camouflages the switch. Flow cytometry data shows the GFP output of the switch. Cells induced with + aTc are represented as red histograms, cells induced with + IPTG are represented as blue histograms. Each stage was performed in triplicate—each histogram represents one biological replicate. Additional histograms at the uncamouflaging stage are negative controls (see Methods for details)

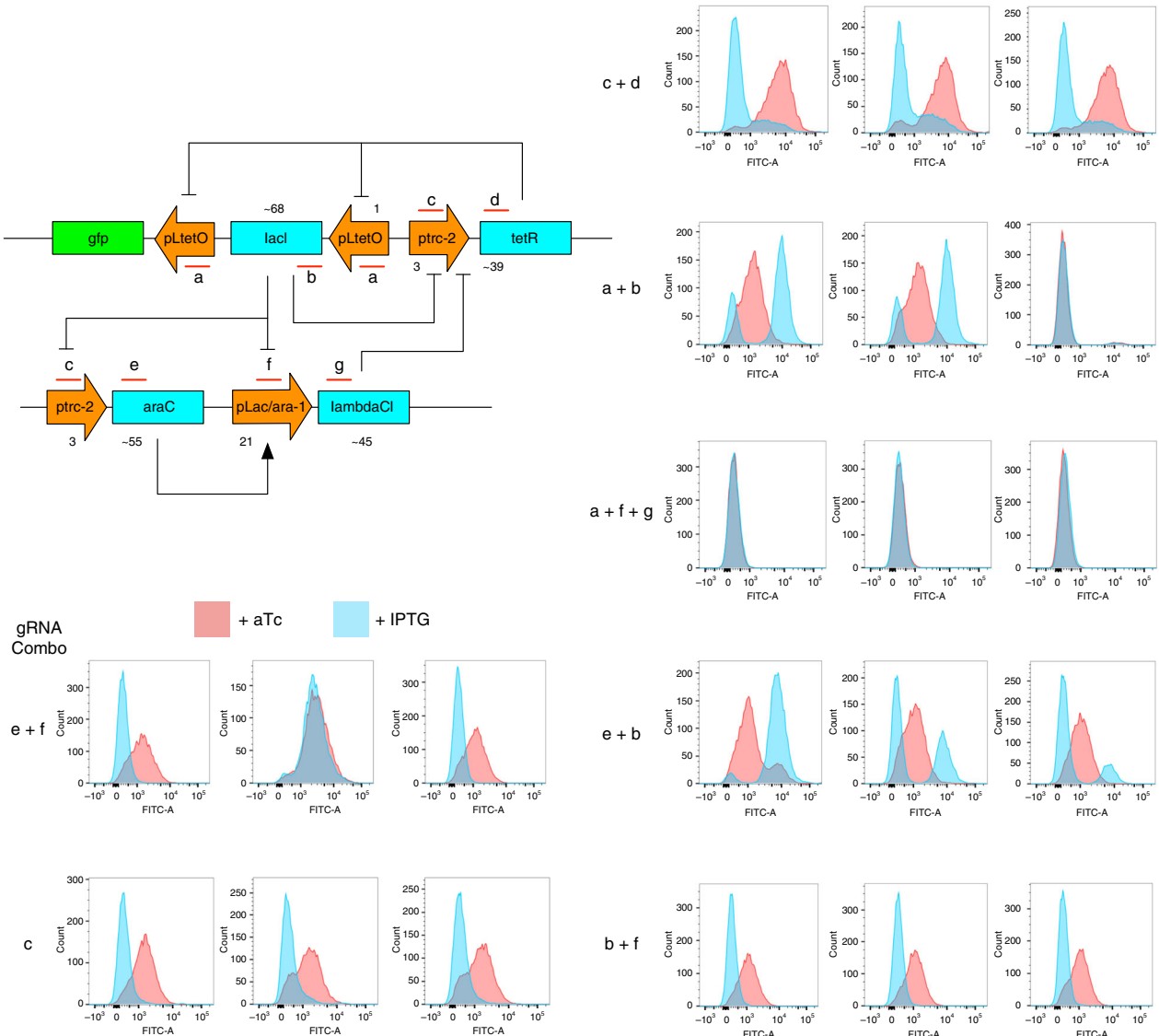

**Fig. 5** Uncamouflaging using incorrect keys. The camouflaged circuit is re-drawn. gRNA targeting positions are shown as a red bar. gRNAs e + g form the correct key, used in Fig. 2. Numbers next to the promoters and ORFs denote the number of potential gRNA-binding sites (i.e., sequences adjacent to an NGG PAM). Histograms show GFP output from incorrect keys. Cell induced with +aTc are represented as red histograms, cells induced with +IPTG are represented as blue histograms. Each experiment was performed in triplicate—each histogram represents one biological replicate

naturally transient, or difficult to detect against the cellular background may minimize this weakness. Examples include directly delivering small RNAs that are typically quickly removed by the cell, or a set of compounds that are quickly metabolized. In each case, without a frequent supply of the key, the circuit would become quickly re-camouflaged.

Traditionally, metabolic engineering has used permanent genetic modification to optimize production of target compounds, such as mutations in specific regulatory sequences or genes to alter their functions. CRISPRi has recently been used to engineer the metabolism of the industrially important *Corynbacterium glutamicium* by repressing genes instead of deleting them[21]. This approach may permit more precise tuning of the host metabolism, allowing configurations of the metabolic network to be achieved that would be difficult to achieve with either mutations or knockouts. Our approach of using a transient decryption key can be applied to metabolic engineering. For any given pathway, there is an optimal flux distribution that maximizes the production of a given target compound. By

transiently adding and removing a dCas9-gRNA plasmid key to modulate the expression of the different pathway components, a strain can be transiently engineered to achieve this optimum. This is an ideal situation whereby the strain only has commercial value when it is de-camouflaged and functioning, while the stored un-engineered base strain would likely have substantially less commercial value.

## Discussion
We have presented two strategies for hiding the design of synthetic gene circuits—circuit scrambling and circuit camouflage, forms of encryption and steganography, respectively—and experimental demonstrations of both concepts. These approaches cover two general strategies: structural modulation of the DNA and targeted modulation of gene expression, respectively. These approaches can also respectively be considered as digital and analog strategies for obfuscation of the circuit, in the sense that structurally rearranging the DNA is all or nothing, while

regulation of gene expression with CRISPRi is generally graded. We demonstrated the strategies of circuit scrambling and circuit camouflage on two different functional circuits, an AND gate and bi-stable switch, respectively. The strategies may have effects on circuit function. For example, our de-camouflaged bi-stable switch displayed different quantitative behaviors compared with the version without any camouflaging genes added. With circuit scrambling, the additional DNA from recombinase sites placed between promoters and ORFs can affect transcription and translation. Furthermore, decoy components that are not excised in circuit scrambling could interact with the real components. Finally, knockdown of camouflaging genes is not absolutely efficient, and thus even weak links between regulators in a circuit may affect the overall circuit performance in subtle ways. These effects are likely to be unnoticeable when only qualitative dynamics are important, but may become apparent if precise quantitative dynamics are required. To prevent this issue, circuits should be optimized in the uncamouflaged state, rather than in the original form.

The utility of circuit scrambling is limited to the scrambling of a circuit to offer protection when the circuit (or a strain engineered with a circuit) is either being stored, or transferred to another party (for instance between collaborators). Circuit camouflage is arguably the superior method as it is currently the simplest to implement and allows for the circuit to be re-hidden, which may be useful for field applications where re-starting from a frozen or dried stock is impractical. The number of combinations is numerically better than circuit scrambling (equal to $2^n$, compared to typically $\leq 2^n$) and will likely result in more plausible (but incorrect) candidates because of an increased number of regulatory links. In our AND gate example, we considered how different candidates could be made to appear more plausible with circuit scrambling, but these techniques are unlikely to apply in the general case and each scrambling will therefore need to be optimized separately. For both circuit scrambling and camouflage, deciding between plausible circuits may be particularly difficult if the encryption/decryption or camouflaging/uncamouflaging results in only a quantitative, not qualitative change. One example is the decryption of the transfer function of a classifier circuit where the topology remains the same, but the "weights" on the links (i.e., the regulatory strength between the genes) are modified. A "quantitative" encryption is harder to implement with circuit scrambling, as different candidates will typically be qualitatively different. For both approaches, creating a list of candidate circuits and then ranking them on the likelihood they are the true circuit would most efficiently be done in silico. Criteria that might be useful for ranking circuits (i.e. predictive of the true circuit) could be: (1) the number and type of logic gates formed in the circuit—particularly for circuits that are suspected to behave digitally and perform complex functions a substantial number of logic gates might be expected, (2) measures of the complexity of the network structure[22], for example if these can be correlated with measures on known circuit designs, and (3) if dynamical models of the circuits are automatically built in silico then the dynamics of the circuit under numerical simulation can be obtained, and a phase space and bifurcation analysis can be performed, all which could be used in some capacity as predictors. In all cases, using prior knowledge of the application domain of the circuit would help in ranking—for example a circuit from a company known to classify disease states might be expected to contain multiple stable equilibria, while one from a company involved in measurement might contain structures that permit more linear dose–responses[23].

There is a third potential approach, re-coding the genetic code, which aims to encrypt a circuit at the level of the genetic code and represents another distinct strategy for encryption.

Circuit scrambling and circuit camouflage offer protection when the circuit is not in use. Conceptually, an optimal scheme would allow for the design of the circuit to remain hidden while the circuit is still functional. Using synthetic genetic codes in which an artificial and secret mapping from codons to amino acids hides the identity of ORFs of the circuit is one approach that could offer this. However, there is a potential weakness to this approach. With circuit scrambling and camouflage, the circuit is non-functional when hidden, but a circuit using a synthetic genetic code would be functional at all times. A third party may only be interested in obtaining the functional cells, rather than also deciphering the design of the circuit that enables the function, and this scenario is a weakness and trade-off for the convenience of not requiring a decryption stage prior to use.

Encryption using circuit re-encoding allows for simultaneous encryption and functioning of the circuit, but is technically challenging to implement currently. Practically, a synthetic genetic code would require orthogonal translation machinery, specifically a set of artificial tRNAs that function only with the codons of the synthetic code, and not with codons of endogenous transcripts. The technology to achieve this is still being developed, but some of the proof-of-principles have been demonstrated;[24–32] Orthogonal ribosomes have already been developed that only recognize and translate corresponding orthogonal mRNAs through a modified Shine–Dalgarno sequence[30], and using quadruplet anti-codon tRNAs that are only incorporated by a mutant orthogonal ribosome[31] is already a viable route to creating an orthogonal tRNA set that would allow for a user-specific quadruplet code for each amino acid.

Biochemically characterizing the orthogonal charged tRNA-amino acids pairs would be possible using modern biochemical techniques and this is arguably the most vulnerable point of the approach. One strategy could be to use a molecular key to repress expression of members of a superset of tRNAs, leaving just the correct set for the circuit. However, this would eliminate the convenience of not requiring a decryption stage.

Ultimately, any encryption scheme is only as strong as its weakest link. For instance, using a recombinase-encrypted circuit, but maintaining the sequence of recombinases to be used for decryption in an unsecured form, or having no security for the plasmid containing the key in circuit camouflage, defeats the purpose of the encryption. Finally, it may be beneficial to combine different types of protection. For instance, storing and transferring an organism in an encrypted form, decrypting when required, and then destroying the key (in the case of circuit camouflage) or the organism's genome, or both, with targeted degradation after its function has been fulfilled[3].

To date, there has only been minimal consideration of how to technologically protect the IP of circuits encoded into engineered cells and organisms. As the value and complexity of these entities increases, the need for hiding the circuit design will also increase. Here, we have presented three conceptually distinct strategies that can form the basis of future work in this direction. We envision that further research into this area, along with efforts to break these systems, will help the field of biological encryption and steganography to advance towards practical implementations.

## Methods

**Strains**. *Escherichia coli* strain MK01[33] was used for the recombinase scrambling experiments in Fig. 1. MK01 is a strain that has shown to give a gradual induction of AraC function with arabinose and also has *lacI* knocked out. *Escherichia coli* strain MK02[33] (equivalent to MK01 but with the chloramphenicol resistance gene removed) was used for the circuit camouflage experiments in Figs. 4, 5. All solid and liquid media was Luria-Bertani (LB). Carbenicillin, kanamycin, and chloramphenicol were used at a final concentration of 50 μg/ml, 30 μg/ml, and 25 μg/ml, respectively. L-arabinose was used at a final concentration of 0.05% (w/v).

**Plasmids**. The encrypted AND gate (pOP437) was constructed by Gibson assembly of multiple synthesized fragments. The plasmid conferred ampicillin resistance and had a pSC101 origin of replication. The (unencrypted) bi-stable switch was obtained from the lab of James Collins (MIT). The stop codons of both *tetR* and *lacI* were changed to TAA, and the existing origin of replication was replaced with the P15A origin, to form pOP512. pOP512 conferred kanamycin resistance. To construct a bi-stable switch capable of encryption by λCI and *araC*, an OR_1-binding site for λCI was inserted between the −10 and −35 regions of the ptrC-2 promoter expressing *tetR*, to form pOP549. λCI and *araC* were cloned into a separate plasmid (to form pOP523) with chloramphenicol resistance and a pBBR1 origin of replication. Decryption plasmids (pOP587–pOP594) were ampicillin resistant and contained the temperature-sensitive origin repA101ts (from pDK46). dCas9 was constitutively expressed from the proD promoter, and gRNAs were constitutively expressed individually and in tandem from the promoter BBa_J23119, and with either the T0 or T1 terminator. dCas9 had no terminator. pOP576 was a control plasmid and only contained proD-dCas9. Correspondence between plasmids and gRNA combinations is as follows: pOP588 −e + f, pOP589 − c, pOP590 − c + d, pOP591 − a + b, pOP592 − a + f + g, pOP593 − e + b, pOP594 − b + f (Fig. 5).

**Recombinases**. Purified recombinase proteins were obtained from BlueSky Bioservices (Worcester, MA). Conditions for recombination were adapted from Ghosh et al.[14]. The buffer comprised 20 mM Tris-HCL pH7.5, 10 mM EDTA, 25 mM NaCl, 10 mM Spermidine, 1 mM DTT (added fresh each time), and 0.1 mg/ml BSA.

**Recombination experiments**. For recombination, typically 1 µl of recombinase was added to 19 µl of recombination buffer, to which ~10 ng of plasmid was then added. Reactions were carried out at either 30 or 37 °C, typically overnight. An aliquot of this mixture was then transformed into *E. coli* DH5alpha cells to propagate the plasmids.

**AND gate characterization**. The decrypted AND gate (Fig. 2, step 1) and the gate at decryption steps 2, 3, and 4 (Fig. 2) correspond to plasmids pOP437, pOP440, pOP447, and pOP458, respectively. Each plasmid was transformed into MK01 cells. Three colonies were picked and grown overnight at 37 °C in LB with carbenecillin. Fresh cultures were inoculated with overnight cultures (1:100 dilution), induced with a final concentration of 1 mM IPTG and arabinose, grown at 37 °C to mid-log and then analyzed on a BD LSRfortessa II Flow cytometer with a 488 nm laser. All histograms are from >5000 cells. Typical gating used is shown in Supplementary Figure 2.

**Bi-stable switch and circuit camouflage experiments**. Bi-stable switch experiments (Figs. 1, 2) were performed as follows: MK02 cells were transformed with pOP512. Three colonies were picked and grown overnight at 30 °C in LB with appropriate selection and arabinose. Fresh cultures inoculated at a 1:1000 dilution were then induced with either aTc or IPTG, at final concentrations of 250 ng/ml and 1 mM respectively, and grown overnight in appropriate selection and arabinose. 500 µl of each culture was then removed and centrifuged at 10×g. The supernatant was then removed, the pellet re-suspended in 500 µl of 1× PBS, and then used to inoculate (1:1000) a fresh culture, without inducer, which was grown overnight at 30 °C in appropriate selection and arabinose. Cells were then analyzed on a BD LSRfortessa II Flow cytometer with a 488 nm laser. All histograms are from >5000 cells. Typical gating used is shown in Supplementary Figure 2

Uncamouflaging was performed as follows: MK02 cells were previously co-transformed with pOP549 and pOP523 to form the camouflaged bi-stable switch strain (opf35). Competent opf35 cells were then transformed by heat shock with either the control (pOP576) or gRNA-containing plasmid (pOP587–pOP594) and grown overnight at 30 °C on solid media with appropriate selection. Data for the camouflaged bi-stable switch was generated from opf35 streaked out from the same competent cell batch used for transformation with the pOP587 decryption plasmid. From each transformation, three colonies were picked and the assay protocol then followed the bi-stable switch protocol described above.

Re-camouflaging continued on from uncamouflaging as follows: cells from each of the three cultures were streaked out on LB plates with kanamycin and chloramphenicol selection and grown overnight at 42 °C. Corresponding mini-streakouts of three colonies from each plate were then made on both LB + kanamycin/chloramphenicol and LB + kanamycin/chloramphenicol/carbenicillin plates. In all cases, the colonies were able to grow on LB + kanamycin/chloramphenicol but not on LB + kanamycin/chloramphenicol/carbenicillin, indicating plasmid loss. One mini-streakout from each plate was then grown overnight at 30 °C in LB with appropriate selection and arabinose. Further steps (induction, washing with PBS, and cytometry) were identical to the uncamouflaging stage.

## Data availability

Raw data for Figs. 2, 4, and 5 is available as a Mendeley data set and can be found at https://doi.org/10.17632/n2h4sfmz4w.1. Annotated plasmid sequences are also available as a Mendeley data set and can be found at https://doi.org/10.17632/88m95bndgm.1.

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

## Acknowledgements

This work was supported by the Defense Advanced Research Projects Agency. MK01 and MK02 cells were kind gifts from M. Kogenaru. The bi-stable switch was gift from J. Collins (MIT). We would like to thank J. W. Purcell for help with experimental work, and N. Roquet and F. Farzafard for reading and helping revise the manuscript.

## Author contributions

O.P., P.S. and T.K.L. conceived the concept of circuit scrambling. J.W. conceived the algorithm for circuit scrambling. O.P. conceived the concepts of circuit camouflage and circuit re-encoding. O.P. conducted experimental work. O.P. and T.K.L. wrote the manuscript.

## Additional information

**Competing interests:** TKL is a co-founder of Senti Biosciences, Synlogic, Engine Biosciences, Tango Therapeutics, Corvium, BiomX, and Eligo Biosciences. TKL also holds financial interests in nest.bio, Ampliphi, IndieBio, and MedicusTek. The remaining authors declare no competing interests.

