## [Peer Review File · Nature Communications]

Reviewers' Comments:

Reviewer #1:

Remarks to the Author:

This is an interesting and thought-provoking paper that lies somewhere in between conceptual and experimental. It will be of interest to the broad scientific community. The experimental work is adequate to show a proof-of-concept, clearly much more work will need to be done in this direction to demonstrate practical applications of this idea. I can recommend publication after the comments below have been addressed:

Specific comments

1.

The authors fail to cite two studies that deal extensively with circuit structure manipulation using recombinases, in particular ref. 2 below. Those should be cited and the relationship discussed between these studies and the current submission

Ref. 1 Lapique et al Nature Chemical Biology 10, 1020–1027 (2014)

Ref. 2 Lapique et al Nature Nanotechnology 13, 309–315 (2018)

2.

The correct unscrambling depends on the order of recombination steps as the authors rightfully mention. It might be difficult to evaluate all the possible permutations experimentally, but it will be trivial to do so in silico. If a rogue agent sequences the scrambled circuit and the recombinase sites are recognized as such by any feature detection software, billions of in silico recombination steps can be done in minutes, and all the resulting circuits can be evaluated in silico whether they "make sense" or not. I would like to see a discussion of this

3.

The discussion of alternative genetic code is not supported by data and it is not clear why it should be mentioned. One can come up with many many additional strategies, so why state this one and not the others? But this is not a major issue, the discussion can be kept as it is.

Reviewer #2:

Remarks to the Author:

The authors proposed two methods to obfuscate the DNA design of a gene circuit, DNA scrambling and transcriptional camouflage. This is done by designing a DNA circuit very dissimilar to the original but where a set of chemical operations could precisely (uniquely) restore its functionality. The more precise the more useful, so I expect the authors will provide some evidence of this (see later my comments).

The goal of encrypting is the one of creating a large diversity in phenotypic space (not necessarily genotypic space), so as many alternative circuits behaviours (all of them wrong except the good one) are produced. Although it is true that phenotypic diversity is not required (only that the phenotype is different from the optimal one), most often the diversity will span many behaviours. This goal is shared in combinatorial approaches to select desired phenotypes. The key difference is on the deterministic way to create diversity. For the scrambling, they use in vitro recombinases to recursively flip/excise the DNA. For the transcriptional camouflage, they add many new transcriptional regulations. The methods for multiplexing recombinases and CRISPRi are now well stabilised, but this new application is interesting. Although it is not clear to me if this method would be the most appropriate (see my comments), it will be of interest for the research up of

Nature Communications provided the authors respond the comments.

Mayor comments:

1. For the Scrambling, the original living system could be considered just as a DNA repository and the recombinase-driven in vitro manipulations they do could be replaced by any other controlled in vitro DNA manipulation. For instance, DNA assembly: We could assemble on demand the circuit by using custom oligos as linkers, which could be designed to select a given assembly instead of many alternative ones. Those linkers could be designed to be orthogonal and it would be much easier than using recombinases. The DNA could come from a living system or directly from gene synthesis. This would be more scalable and simpler, or not?
2. Could the authors comment on the use of their scrambling as a way to create libraries of randomised sequences? If the recombinases are expressed in vivo, this could also be used as a memory system. Comment on the cre/loxP randomization to color neurons paper.
3. The authors need to check for off-target/cross-talk recombinations that could give the wrong sequence. The authors should provide quantitative evidence of how much of the correct sequence is obtained versus the wrong ones. The more scrambling layers are used the larger will be the generation of wrong sequences. After the scrambling operations, in general a library of alternate DNAs could be produced because of off-target/cross-talk effects in the enzymes. In fact already the author's estimate is of around 40%. With sequencing it is easy to get the real number. The authors need to quantify it.
4. Still is not possible to develop a general methodology for the directed evolution of gene circuits, but in many cases this is possible. Could any hacker "decompile" the scrambling/transcriptional obfuscation by simply doing a selection process? Directed evolution can often overcome large sequence spaces because there is often some adaptation (as it already happens to the authors).
5. Could the authors provide an estimation of the minimum entropy or at least Shannon entropy to quantify the degree of obfuscation? Could an experimental quantification of circuit behaviour be related to such measure?
6. Could the authors discuss about the possibility of combining both approaches? In particular if recombinases are used in vivo as part of the circuit, their regulation with CRISPRi would add extra obfuscation. Could the authors do a theoretical estimation of this?
7. The use of recombinases could be considered as a digital (Boolean) form of obfuscating a circuit and the addition of spurious transcription regulations as an analog way of obfuscating. The authors already imply this fact during the discussion of the obfuscation of a metabolic pathway, but I recommend to explicitly state it because it may be inspiring to the readers.
8. As an alternative, what happens if we use exogenous gRNAs to implement the circuit (in the cases this is possible)? Would not this be simpler than adding many fake transcriptional regulators to later silence them?

Minor comments:

1. Fig. 1A: aTc should be arabinose, also instead of a repression it should be activation in the araC and pLac/ara1 regulations.
2. Fig 2: the same, replace aTc by arabinose
3. Annotated sequences should be provided for each plasmid

Response to reviewers.

Reviewer 1.

1. The authors fail to cite two studies that deal extensively with circuit structure manipulation using recombinases, in particular ref. 2 below. Those should be cited and the relationship discussed between these studies and the current submission

Ref. 1 Lapique et al Nature Chemical Biology 10, 1020–1027 (2014)

Ref. 2 Lapique et al Nature Nanotechnology 13, 309–315 (2018)

We thank the reviewer for highlighting these two relevant papers. We have included their discussion and references to them as both examples of the use of order dependency of uni-directional recombinases in genetic circuit design and the interesting use of bi-directional recombinases to “decompress” a single genetic circuit structure and generate an equilibrium of more than one circuit structure. This is added on lines 97-101.

2. The correct unscrambling depends on the order of recombination steps as the authors rightfully mention. It might be difficult to evaluate all the possible permutations experimentally, but it will be trivial to do so in silico. If a rogue agent sequences the scrambled circuit and the recombinase sites are recognized as such by any feature detection software, billions of in silico recombination steps can be done in minutes, and all the resulting circuits can be evaluated in silico whether they "make sense" or not. I would like to see a discussion of this

We thank the reviewer for highlighting an area that we had not fully discussed. Although we had discussed ways to make the determination of the true circuit more difficult, we had not discussed criteria that could be used to rank a list of possible candidates. Although the reviewer only mentions this in the context of circuit scrambling the same is true for circuit camouflage, so we have added a discussion in the final discussion section.

We have added text discussing this on lines 421-435.

3. The discussion of alternative genetic code is not supported by data and it is not clear why it should be mentioned. One can come up with many many additional strategies, so why state this one and not the others? But

this is not a major issue, the discussion can be kept as it is.

The reviewer is right that there are additional strategies that could be conceived. We chose to include the discussion of alternative genetic codes, as it allows us to make the discussion more comprehensive by covering a third conceptual approach. Circuit scrambling rearranges the physical structure of the DNA that forms the circuit. Circuit camouflage keeps the DNA that forms the circuit unchanged, but hides it in additional circuit structures. In contrast the use of an alternative genetic code requires no change or addition to the circuit DNA. Furthermore, as discussed it allows the circuit to remain functional when it is encrypted, a notable difference from circuit scrambling and camouflage. We feel that it's discussion helps to put the other two approaches in context and we are pleased that the reviewer is happy for us to keep this discussion in the paper.

Reviewer 2.

The authors proposed two methods to obfuscate the DNA design of a gene circuit, DNA scrambling and transcriptional camouflage. This is done by designing a DNA circuit very dissimilar to the original but where a set of chemical operations could precisely (uniquely) restore its functionality. The more precise the more useful, so I expect the authors will provide some evidence of this (see later my comments).

The goal of encrypting is the one of creating a large diversity in phenotypic space (not necessarily genotypic space), so as many alternative circuits behaviours (all of them wrong except the good one) are produced.

Although it is true that phenotypic diversity is not required (only that the phenotype is different from the optimal one), most often the diversity will span many behaviours. This goal is shared in combinatorial approaches to select desired phenotypes. The key difference is on the deterministic way to create diversity. For the scrambling, they use in vitro recombinases to recursively flip/excise the DNA. For the transcriptional camouflage, they add many new transcriptional regulations. The methods for multiplexing recombinases and CRISPRi are now well stabilised, but this new application is interesting. Although it is not clear to me if this method would be the most appropriate (see my comments), it will be of interest for the research up of Nature Communications provided the authors respond the comments.

Major comments:

- 1. For the Scrambling, the original living system could be considered just as a DNA repository and the recombinase-driven in vitro manipulations they do could be replaced by any other controlled in vitro DNA manipulation. For instance, DNA assembly: We could assemble on demand the circuit by using custom oligos as linkers,**

which could be designed to select a given assembly instead of many alternative ones. Those linkers could be designed to be orthogonal and it would be much easier than using recombinases. The DNA could come from a living system or directly from gene synthesis. This would be more scalable and simpler, or not?

The reviewer makes a good point and it is true there are other levels on which a genetic circuit could be “hidden” – in the reviewers example the circuit is never constructed until the oligo “key” is supplied allowing the correct circuit to be assembled. You can go a step further and simply encrypt the DNA sequence of the circuit on a computer, which would be encrypted/decrypted using standard approaches and which would then be sent to an automated DNA construction pipeline to be built.

The main consideration here is the practicality of assembling a circuit every time it needs to be used. In the majority of cases, for the majority of groups, and in particular for larger circuits, circuit synthesis and assembly is still an inefficient process. Whether it is more or less efficient that the circuit scrambling approach is not clear as the two approaches would both need to be optimized and then compared. My feeling is that currently the reviewers approach would be more efficient/simpler for smaller circuits, while scrambling would be more efficient and simpler for larger circuits.

Another consideration is the introduction of the circuit into a cell. In this paper we performed all recombinase reactions in vitro. However decrypting the circuit in vivo is possible, by sequentially transforming in DNA encoding the recombinases. For larger circuits in difficult to transform cell types this would allow the circuit to be introduced once, and then decrypted when it needed to be used, in contrast to an approach whereby the circuit needed to be assembled and transformed every time.

- 2. Could the authors comment on the use of their scrambling as a way to create libraries of randomised sequences? If the recombinases are expressed in vivo, this could also be used as a memory system. Comment on the cre/loxP randomization to color neurons paper.**

I think one could conceive of taking a set of recombinase recognition sites and placing them around a number of components and then treating the mixture with all those recombinases to generate diversity – I’m sure this is being done in some context by certain groups, if not already published. I don’t think this is related in particular to the scrambling approach as scrambling relies on a single route for recombinase action using recombinases applied sequentially. However, if you did generate a high sequence diversity library with a set of recombinases in vivo then this could be used as an alternative to the colored approach described in that

paper. The obvious benefit of the colored approach is that it can easily be visualized, whereas sequencing would be needed to understand the lineage otherwise.

- 3. The authors need to check for off-target/cross-talk recombinations that could give the wrong sequence. The authors should provide quantitative evidence of how much of the correct sequence is obtained versus the wrong ones. The more scrambling layers are used the larger will be the generation of wrong sequences. After the scrambling operations, in general a library of alternate DNAs could be produced because of off-target/cross-talk effects in the enzymes. In fact already the author's estimate is of around 40%. With sequencing it is easy to get the real number. The authors need to quantify it.**

We agree that off-target effects of recombinases are an important consideration, and we thank the reviewer for bringing this to our attention. However, the off target activities of set of 11 different recombinases have been extensively quantified elsewhere and are discussed in the manuscript (Lines 177-180). When using circuit scrambling it would be simplest to just use this data to pick a set which shows maximal orthogonality/minimal cross-talk. As such while the particular set we used for our proof-of-concept have not yet been compared all together, the omission of orthogonality data for that particular set does not affect the validity of the concept we are conveying. However, we had not included a discussion of the need to consider cross-talk and pick recombinases demonstrating high orthogonality and so we have added this discussion to the manuscript at lines 180-184.

- 4. Still is not possible to develop a general methodology for the directed evolution of gene circuits, but in many cases this is possible. Could any hacker "decompile" the scrambling/transcriptional obfuscation by simply doing a selection process? Directed evolution can often overcome large sequence spaces because there is often some adaptation (as it already happens to the authors).**

It is difficult to conceive how selection could be used to circumvent the obfuscation as it is not clear what the selection criteria would be – the correct behavior to select towards is unknown. Perhaps if an attacker speculated on the purpose of a circuit then used all possible recombinases and selected for circuits that took steps towards this behavior, that might be one approach, but I don't think it would be successful. In this case it would be simpler to use the approach

we discuss of enumerating all possible states of the circuits (likely *in silico*) and then using the prior knowledge of the purpose of the circuit to rank the set of circuits based on their dynamics and function.

5. Could the authors provide an estimation of the minimum entropy or at least Shannon entropy to quantify the degree of obfuscation? Could an experimental quantification of circuit behaviour be related to such measure?

This is an interesting idea but this level of quantification is really beyond the intended scope of this paper – it would be better dealt with in a separate paper specifically dealing with quantifying the strength of these types of encryption schemes. I think in addition to the entropy of simply the number of possible DNA states, which may be possible to calculate (or bound), the important task of estimating the entropy beyond that i.e. the probability that the resulting state is the correct one, is much harder, and I don't think it is simple enough to do that it can be an additional to this work.

6. Could the authors discuss about the possibility of combining both approaches? In particular if recombinases are used in vivo as part of the circuit, their regulation with CRISPRi would add extra obfuscation. Could the authors do a theoretical estimation of this?

We agree that given that different obfuscation approaches will have different weaknesses, a carefully considered combination of approaches might mitigate against any single weakness. We already discuss the idea of combining different approaches (lines 472-475).

With regard to the specific suggestion of using recombinases in vivo as part of the circuit and then regulating them with CRISPRi, I think this is in principle a valid approach, and may add to the level of obfuscation. However, using recombinases in a circuit requires very tight control of the recombinase expression – this is because the action of the recombinase on its uni-directional sites increases monotonically, and even low levels of expression will over time lead to all sites being flipped. CRISPRi is arguably not yet at a level where it is easy to obtain the level of repression that would be required for regulation of recombinase expression.

7. The use of recombinases could be considered as a digital (Boolean) form of obfuscating a circuit and the addition of spurious

transcription regulations as an analog way of obfuscating. The authors already imply this fact during the discussion of the obfuscation of a metabolic pathway, but I recommend to explicitly state it because it may be inspiring to the readers.

We thank the reviewer for pointing this out. It is an interesting and useful way of looking at the differences between these two approaches. We have added this to the discussion at lines 386-389.

8. As an alternative, what happens if we use exogenous gRNAs to implement the circuit (in the cases this is possible)? Would not this be simpler than adding many fake transcriptional regulators to later silence them?

It is not typical to create gene circuits in vivo by addition of only the functional components (i.e. proteins, or RNA/gRNA etc) but not the DNA encoding them – the problem is that it would 1) be difficult, in particular for complex circuits, to get all the components into all the cells (while the DNA encoding them is typically introduced as a single molecule, or sequentially as a small number of different molecules, in either case being stably maintained with selection), and 2) the circuit would only exist for a short time until the components had degraded, as there is no production from genes as is the case when DNA encoding the components is put into the cell. For many applications, longer-term engineering of the cell with the circuit is required (for instance a therapeutic application) and so this might be an issue.

However, it is an interesting idea, that instead of obfuscating a circuit because it exists in the cell long-term (which creates chances for it to be stolen), someone can just create a very transient circuit that exists for a short time, reducing the chance it's identity can be discovered.

Minor comments:

- 1. Fig. 1A: aTc should be arabinose, also instead of a repression it should be activation in the araC and pLac/ara1 regulations.**

We thank the reviewer for highlighting this error, it has been corrected. We also realized the araC repression error is also on figures 4 and 5 and so these have also been changed.

- 2. Fig 2: the same, replace aTc by arabinose**

We thank the reviewer for highlighting this error, it has been corrected.

- 3. Annotated sequences should be provided for each plasmid**

The annotated sequences have been provided as a Mendeley online dataset, available at the following DOI: [10.17632/88m95bndgm.1](https://doi.org/10.17632/88m95bndgm.1)

Reviewers' Comments:

Reviewer #2:

Remarks to the Author:

I am fully satisfied with the authors' replies to all my 8 comments and I recommend the publication of the manuscript as it is in Nature Comms, nice piece of work.

Alfonso Jaramillo